# Antimicrobial Dispensing Practices during COVID-19 and the Implications for Pakistan

**DOI:** 10.3390/antibiotics12061018

**Published:** 2023-06-06

**Authors:** Bushra Gul, Maria Sana, Aneela Saleem, Zia Ul Mustafa, Muhammad Salman, Yusra Habib Khan, Tauqeer Hussain Mallhi, Tiyani Milta Sono, Johanna C. Meyer, Brian B. Godman

**Affiliations:** 1Department of Medicines, Tehsil Head Quarter (THQ) Hospital, District Bhakkar, Darya Khan 3000, Punjab, Pakistan; bushragul434@gmail.com; 2Department of Medicine, Faisalabad Medical University, Faisalabad 38000, Punjab, Pakistan; mariasana04@gmail.com (M.S.); aneelasaleem23@hotmail.com (A.S.); 3Discipline of Clinical Pharmacy, School of Pharmaceutical Sciences, Universiti Sains Malaysia, Gelugor 11800, Penang, Malaysia; 4Department of Pharmacy Services, District Headquarter (DHQ) Hospital, Pakpattan 57400, Punja, Pakistan; 5Institute of Pharmacy, Faculty of Pharmaceutical and Allied Health Sciences, Lahore College for Women University, Lahore 54000, Punja, Pakistan; dr.muhammadsalman@lcwu.edu.pk; 6Department of Clinical Pharmacy, College of Pharmacy, Jouf University, Sakaka 72388, Saudi Arabia; yhkhan@ju.edu.sa (Y.H.K.); thhussain@ju.edu.sa (T.H.M.); 7Department of Public Health Pharmacy and Management, School of Pharmacy, Sefako Makgatho Health Sciences University, Ga-Rankuwa 0208, Gauteng, South Africa; tiyanim@gmail.com (T.M.S.); hannelie.meyer@smu.ac.za (J.C.M.); 8Saselamani Pharmacy, Saselamani 0928, Limpopo, South Africa; 9South African Vaccination and Immunisation Centre, Sefako Makgatho Health Sciences University, Ga-Rankuwa 0208, Gauteng, South Africa; 10Department of Pharmacoepidemiology, Strathclyde Institute of Pharmacy and Biomedical Science (SIPBS), University of Strathclyde, Glasgow G4 0RE, UK; 11Centre of Medical and Bio-Allied Health Sciences Research, Ajman University, Ajman P.O. Box 346, United Arab Emirates

**Keywords:** COVID-19, dispensing, antimicrobials, antibiotics, community pharmacists, indications, AWaRe category, Pakistan

## Abstract

Antibiotics are one of the most frequently dispensed classes of medicines. However, excessive misuse and abuse enhances antimicrobial resistance (AMR). Previous studies in Pakistan have documented extensive dispensing of ‘Watch’ and ‘Reserve’ antibiotics, which is a concern. In view of this, there is a need to assess current dispensing patterns following COVID-19 in Pakistan. A cross-sectional study was undertaken, collecting dispensing data from 39 pharmacies and 53 drug stores from November 2022 to February 2023. Outlets were principally in urban areas (60.9%), with pharmacists/pharmacy technicians present in 32.6% of outlets. In total, 11,092 prescriptions were analyzed; 67.1% of patients were supplied at least one antimicrobial, 74.3% antibiotics, 10.2% antifungals and 7.9% anthelmintics. A total of 33.2% of antimicrobials were supplied without a prescription. Common indications for dispensed antibiotics were respiratory (34.3%) and gastrointestinal (16.8%) infections, which can be self-limiting. In addition, 12% of antibiotics were dispensed for the prevention or treatment of COVID-19. The most frequent antibiotics dispensed were ceftriaxone (18.4%) and amoxicillin (15.4%). Overall, 59.2% antibiotics were ‘Watch’ antibiotics, followed by ‘Access’ (40.3%) and ‘Reserve’ (0.5%) antibiotics. Of the total antibiotics dispensed for treating COVID-19, 68.3% were ‘Watch’ and 31.7% ‘Access’. Overall, there appeared to be an appreciable number of antibiotics dispensed during the recent pandemic, including for patients with COVID-19, alongside generally extensive dispensing of ‘Watch’ antibiotics. This needs to be urgently addressed with appropriate programs among pharmacists/pharmacy technicians to reduce AMR.

## 1. Introduction

Since the emergence of the COVID-19 outbreak caused by the SARS-Co-V2 outbreak in December 2019, neighboring countries to China, including Pakistan, have been at considerable risk of the virus [1,2]. This is enhanced by a porous border, as well as extensive trade and travel ties with China through land, sea and air [2,3,4]. In Pakistan, the first positive case of COVID-19 was reported on 26 February 2020 [4]. This was followed by a substantial number of positive cases throughout the country in various waves of the disease; however, prevalence rates may be under-reported [4,5,6]. Similar to other countries, Pakistan implemented many precautionary measures to slow the spread of the virus and its impact, including country-wide lockdown measures [7,8,9,10]. However, despite these measures, COVID-19 still caused a considerable impact on the country, which included the healthcare system [4,8,9,10]. Up to mid-to-late March 2023, more than 1.57 million positive cases and 30,000 deaths were reported in Pakistan [11]. In late March, more than 3500 suspect patients were being tested for COVID-19 daily, with approximately 100 patients daily being reported to have COVID-19 [12]. In addition to the instigation of multiple preventive measures, certain public sector hospitals were designated for the treatment of patients with presumed moderate-to-severe COVID-19 [13,14]. Many medicines, including corticosteroids, antipyretics, antihistamines and anti-thrombotics, have been prescribed to patients admitted with COVID-19 to different secondary and tertiary hospitals in Pakistan [15,16,17,18]. Alongside this, there was appreciable prescribing of antibiotics, especially ‘Watch’ antibiotics, among patients admitted with COVID-19 to public sector facilities in Pakistan despite limited prevalence of bacterial co-infections and secondary bacterial infections [13,14,18,19,20]. During the first five waves, this averaged 1.14% of patients with bacterial co-infections being prescribed antibiotics and 3.14% with secondary infections, which is similar to other countries [14,21,22,23]. Alongside this, extensive purchasing of antibiotics in Pakistan without a prescription, including both ‘Watch’ and ‘Reserve’ antibiotics, also occurred before and during the current pandemic [24,25,26,27,28]. We have seen appreciable purchasing of antibiotics without a prescription across low- and middle-income countries (LMICs), which includes African and Asian countries, despite high rates of antimicrobial resistance [25,29,30,31,32,33,34]. This is driven by patient requests, previous experiences, limited governance of laws banning such behavior and limited knowledge regarding antibiotics and AMR among key stakeholder groups [34,35,36,37,38,39]. 

This overuse of antibiotics across sectors, particularly ‘Watch’ antibiotics, exacerbated by the recent COVID-19 pandemic, is a concern, as this will increase antimicrobial resistance (AMR) [14,40,41,42], which is already a key issue in Pakistan [43]. Globally in 2019, there were 1.27 million deaths directly attributable to AMR, with this figure expected to continue rising unless addressed [29,44]. There is currently a disproportionate burden of AMR among LMICs, exacerbated by numerous financial, political and sociological factors [29,45,46]. The lack of adequate sanitation and hygiene, as well as poor infection control and preventive practices, alongside excessive and irrational use of antimicrobial agents, have increased the prevalence of AMR in LMICs in recent years [45,47,48]. Pakistan is no exception. 

Pakistan is a LMIC located in South Asia and is currently the third-largest consumer of antibiotics globally, increasing AMR [49,50]. Multi-drug-resistant (MDR) and extensively drug-resistant (XDR) organisms, as well as other resistant organisms, have been reported throughout Pakistan, adding to concerns regarding AMR [50,51]. In view of increasing rates of AMR globally, international organizations, including the World Health Organization (WHO), have instigated a number of international activities. These include the development of the ‘global action plan’ (GAP) against AMR [52,53,54,55]. In line with the recommendations, Pakistan developed its own ‘National Action Plan’ (NAP) against AMR in 2017; however, there are challenges in its implementation [43,56]. 

In both the GAP and the NAP of Pakistan, assessing current antibiotic utilization patterns, alongside issues of inappropriate use, are seen as necessary starting points to develop appropriate plans to reduce AMR. Potential NAP activities include developing pertinent strategies to reduce inappropriate prescribing and dispensing in patients with COVID-19. We are aware that a number of studies have documented inappropriate prescribing and dispensing of antibiotics among patients with COVID-19 in Pakistan [14,19,20,57]. However, we are currently unaware of any study to date that has evaluated dispensing practices for antimicrobials from drug sale outlets during the current COVID-19 pandemic. This is important given growing concerns with increased antibiotic utilization during the pandemic, especially ‘Watch’ antibiotics, and the implications for increasing AMR. Consequently, the aim of this study was to undertake a multicenter study to access current patterns of antimicrobials dispensed in Pakistan at drug sale outlets during the pandemic. The findings can be used to develop future strategies, if pertinent, to reduce rising AMR rates in Pakistan. This can include the instigation of appropriate antimicrobial stewardship programs (ASPs) in ambulatory care to improve future prescribing and dispensing [48,58,59]. 

## 2. Results

Ninety-two drug sale outlets were included in the study, which incorporated 53 medical stores and 39 pharmacies (Table 1). Most of the sales outlets were located in urban areas (60.9%), with only one-third (32.6%) of outlets having a qualified pharmacist or pharmacy technician dispensing medicines. Most of the patients who were dispensed antibiotics were male (55.2%) and aged 46–60 years (29.5%) out of a total of 11,092 encounters where patients were dispensed a medicine. 

More than two-thirds of the encounters (67.1%) resulted in antimicrobials being dispensed (Table 2). Among the dispensed antimicrobials, nearly three-quarters (74.3%) were antibiotics, followed by antifungals (10.2%) and anthelmintics (7.9%). In total, 48.7% of antimicrobials dispensed were oral preparations, with 33.2% of antimicrobials being dispensed without a prescription from registered medical practitioners. 

The indications for the dispensed antimicrobials are given in Table 3. As can be seen, more than a quarter (34.3%) of the antibiotics dispensed were for respiratory tract infections, followed by gastrointestinal (16.8%), skin and soft-tissue infections (12.9%), as well as pre- or post-operative prophylaxis (12.6%). Out of the total number of antibiotics dispensed in the four divisions, 12% of antibiotics were dispensed for patients with COVID-19. 

Details of the top ten most frequently dispensed antibiotics are shown in Table 4. The most frequently dispensed antibiotics were ceftriaxone (18.4%—‘Watch’), amoxicillin (15.4%—‘Access’) and azithromycin (14.9%—‘Watch’). 

The most frequently dispensed antibiotics for patients with COVID-19 were ceftriaxone (28%—‘Watch’), followed by azithromycin (25%—‘Watch’) and amoxicillin (22.4%—‘Access’) (Table 5).

Antibiotic dispensing practices as per the WHO AWaRe classification are shown for each Division in Figure 1. As can be seen, most of the antibiotics dispensed were from the ‘Watch’ category (59.2%) followed by the ‘Access’ category (40.3%), with a few also from the ‘Reserve’ category (0.5%). 

Dispensing practices of antibiotics among patients with COVID-19 showed that slightly more than two-thirds of antibiotics dispensed (68.3%) were from the ‘Watch’ category, followed by the ‘Access’ category (31.7%). No antibiotic dispensed was from the ‘Reserve’ category (Figure 2).

## 3. Discussion

We believe this is one of the first studies from Pakistan that has assessed antimicrobial dispensing practices during the current COVID-19 pandemic. This is important given concerns across countries with the over-prescribing of antibiotics in patients with COVID-19 despite limited evidence of bacterial infections or co-infections [21,22,23,62,63,64]. In addition, extensive prescribing of ‘Watch’ antibiotics has been seen in patients hospitalized with COVID-19 in Pakistan [13,14,20]. 

More than two-thirds of the total patient encounters in our study were supplied with at least one antimicrobial, with nearly three-quarters of these being antibiotics. This overuse of antibiotics in ambulatory care during the COVID-19 pandemic is in line with previous studies from other South Asian countries and beyond [62,63,64,65]. A recent study analyzing sales data from 71 countries also concluded similar findings, with an increase in antibiotic utilization during the COVID-19 pandemic [66]. 

This is a concern alongside the finding that respiratory tract infections and gastrointestinal infections were the most common complaints and where antibiotics are typically not necessary [67]. Others common indications were the prevention and treatment of COVID-19, where again antibiotics may not be necessary, with, as mentioned, little evidence of bacterial infections in these patients [14,22,23]. This is similar to a study from Russia where the majority of the antibiotics dispensed in ambulatory care were for respiratory and skin and soft-tissue infections [68].

The top three antibiotics dispensed at the surveyed drug sale outlets in Pakistan during the study period were ceftriaxone, amoxicillin and azithromycin. The appreciable use of ceftriaxone and azithromycin resulted in most of the antibiotics dispensed during the study period being from the ‘Watch’ category (59.2%), followed by the ‘Access’ category (40.3%), in addition to a minority from the ‘Reserve’ group of antibiotics (0.5%). This mirrors other studies in Pakistan pre-pandemic, where there was appreciable dispensing of ‘Watch’ and ‘Reserve’ antibiotics [24], as well as in Bangladesh [69]. This high use of ‘Watch’ antibiotics alongside any dispensing of ‘Reserve’ antibiotics needs to be urgently addressed if Pakistan is to reduce current high rates of AMR in line with the objectives of the NAP [43].

Similarly, patients with COVID-19 were frequently dispensed ceftriaxone, azithromycin, amoxicillin and ciprofloxacin. As a result, 68.3% of antibiotics dispensed for these patients were from the ‘Watch’ category, higher than those without COVID-19. Whilst our findings are comparable to those from Egypt, where ceftriaxone and azithromycin were again common antibiotics dispensed during COVID-19 [70], this also needs to be urgently addressed.

Of equal concern is that one-third of the total antibiotics dispensed by the surveyed outlets in our study were without a prescription from registered medical practitioners. Whilst this is an improvement from previous studies in the Hazara Division of Khyber Pakhtunkhwa Province, Pakistan, where more than 90% antibiotics were dispensed without prescriptions [71], as well as Lahore, where more than 95% of antibiotics were dispensed without a prescription [24], it is still against current regulations [72]. This is important for respiratory tract infections and gastrointestinal infections, common complaints where antibiotics were dispensed, for which antibiotics are often inappropriate [67]. 

We have seen a number of antimicrobial stewardship programs (ASPs) introduced in ambulatory care across LMICs to improve appropriate prescribing. These are described in Appendix A. There was a concern that ASPs were difficult to undertake in LMICs due to personnel and financial issues [73]. However, this no longer appears to be the case, with a number of exemplars discussed in Appendix A providing direction to all key stakeholder groups in Pakistan. Active follow-up of any intervention is essential for the sustainability of any ASP (Appendix A). We are also aware of studies in other countries where ASPs have been successfully introduced in community pharmacies to reduce inappropriate dispensing of antibiotics (Appendix A). Alongside this, initiatives in LMICs that have been introduced but have failed to sustainably reduce inappropriate dispensing of antibiotics without a prescription. These latter initiatives typically failed due to a lack of follow-up and monitoring of community pharmacists’ activities. Consequently, any initiative introduced in Pakistan needs to be actively followed up or else the objectives will not be met. This is because it is essential that inappropriate dispensing of antibiotics is addressed to reduce AMR. Community pharmacists and pharmacy technicians will play a greater role in improving patient care across countries in the future, including LMICs, building on their increasing role during the COVID-19 pandemic [74,75,76,77,78]. In view of this, an increasingly important target group to improve antibiotic utilization is in ambulatory care. However, there are typically knowledge gaps among community pharmacists and pharmacy technicians regarding antibiotics and AMR in LMICs which urgently need to be addressed [39,79,80]. This starts in universities and other educational establishments, and continues post-qualification for maximum impact given current knowledge concerns among pharmacists and pharmacy technicians in Pakistan [79,81,82]. We have seen trained pharmacists, coupled with the use of guidelines suggesting alternatives to antibiotics, being used to good effect in LMICs to limit the dispensing of antibiotics for essentially self-limiting viral infections [78,83,84], and as a result providing future guidance to key stakeholder groups in Pakistan [78,83,84].

Another concern is that many of the drug outlets in our study did not have qualified pharmacists or pharmacy technicians to help give alternative advice for self-limiting conditions such as URTIs. Whilst this is similar to other countries [85], this also needs to be addressed in Pakistan going forward. In the first instance, this includes educational programs among all key dispensing personnel, along with guidelines and targets based on the AWaRe book with its multiple guidance on key infections in ambulatory care, to reduce inappropriate dispensing [67,83,84]. This is seen as more likely to succeed than introducing limited fines for community pharmacists for dispensing antibiotics without a prescription, as seen in Vietnam (Appendix A) [86]. 

We are aware of a number of limitations associated with our study. Firstly, we only conducted this study in four divisions of Punjab Province. However, we chose Punjab Province for this initial study for the reasons documented. In addition, we included an appreciable number of drug outlets and patients to help address concerns with bias. We also did not collect information regarding the signs and symptoms and laboratory findings of patients to fully assess the appropriateness of dispensed antibiotics. However, this was impractical given that the principal objective of this study was to assess actual antimicrobial dispensing practices. In addition, we only collected the number of antibiotics dispensed for particular conditions rather than the extent of multiple, as opposed to single, antibiotics for given patients for identified indications, as our principal objective was again to assess the actual extent of antibiotics dispensed for common infections. We will, however, be following this up in future studies. Despite these limitations, we believe our findings are robust and can provide guidance to health authorities, healthcare providers, including those working in drug sales outlets, as well as the general population in Pakistan, to reduce high rates of inappropriate dispensing of antibiotics in patients with or without COVID-19. This is important as Pakistan strives to reduce its high rate of AMR as part of its agreed NAP.

## 4. Materials and Methods

### 4.1. Study Design and Setting

This cross-sectional study was conducted in drug sale points (pharmacies/medical stores) in four divisions of the Punjab Province to evaluate the dispensing practices of antimicrobial agents during the current COVID-19 pandemic. Pharmacies are drug sale point/premises working under the supervision of a qualified pharmacist, i.e., a graduate in pharmacy which takes five years. Medical stores are run by pharmacy technicians (two-year diploma in pharmacy).

There are 11 divisions in Punjab and we conveniently selected 4 of these in this study, subsequently named D1 to D4. We chose the Punjab Province for this initial study as it covers more than half of the population of Pakistan; consequently, the findings can give good insight into the remainder of Pakistan [20].

Subsequent pharmacies and medical stores were conveniently selected to collect dispensing information in line with other studies. This included previous studies involving the co-authors [79,87,88]. A minimum of five pharmacies and ten medical stores were included from every division, with a minimum of 100 prescriptions included from each pharmacy and 50 from each medical store. There was no sample size calculation as this type of research has not been undertaken before in Pakistan among patients with COVID-19. However, we aimed for a substantial number of prescriptions and data to add robustness to our findings.

As per the drug law in Pakistan of 1976, medicines can only be sold at medical stores and pharmacies, and only following a prescription [72]. Medical stores are abundant, particularly in rural areas, where medical stores can be operated under the supervision of pharmacy technicians/dispensers. However, pharmacies are usually located in urban areas and these outlets work under the supervision of a pharmacist. The drug outlets provided medicines, including antimicrobials, vaccines and personal protective equipment (PPE), during the recent COVID-19 pandemic.

### 4.2. Study Variables

Information about the drug sale outlets included in the study, including pharmacies and medical stores, was collected by the investigators. This included their numbers, location, presence of qualified personnel at the pharmacy/medical store and whether the drug sales point was working independently or as a branch of a chain pharmacy/medical store. 

The total number of patient encounters recorded and the encounters dispensed with at least one antimicrobial agent were recorded. The gender and age of the patients were documented alongside the type of antimicrobial dispensed. This included whether this was an antibiotic, antiviral, antifungal, antiprotozoal or anthelmintic. The route of administration was also documented.

Information about the number of encounters was collected by the investigators. We included all encounters recorded during the data collection period. We also included those encounters that included other medicines being dispensed alongside antibiotics. In addition, where antibiotics were being demanded from drug outlets without a prescription. 

The dispensing practices surrounding antimicrobials were also recorded. This included the number of prescriptions that were dispensed with at least one antimicrobial out of total number of encounters, alongside the generic name of the antimicrobial (International Nonproprietary Name—INN [89]), the route of administration, the indications for any antibiotics dispensed as well as the type of antibiotic dispensed according to AWaRe classification [60,61]. Antibiotics dispensed for positive or suspected COVID-19 patients were also recorded alongside their categorization as per the AWaRe classification. 

### 4.3. Data Collection Process

Investigators, including pharmacists and pharmacy technicians, visited drug sale outlets during the study period and requested the proprietor/owner or community pharmacist/pharmacy technicians of the concerned drug sale point to allow them to collect dispensing data after describing the study objectives. Those who were ready to facilitate this survey were briefed about the data collection process. Investigators were available at the drug sale point where medicines were being dispended and patients were counselled about the objectives of the study. Written informed consent was obtained from potential patient contributors prior to their enrollment into the study. No personal information, including the name of possible patients, their telephone number or national identity card number, was recorded to maintain confidentiality. 

### 4.4. Statistical Analysis

The SPSS version 22.0 for Windows and Microsoft Excel version 2016 were used for analyzing the data of this study. Numbers and percentages were used to present categorical variables. Participants’ data were coded and stored in a password-protected file accessible only to the researchers to maintain confidentiality.

### 4.5. Study Approval

This study was approved by the ORIC, Lahore College for Women University (LCWU) Lahore, Pakistan. Written informed consent was obtained from all the study participants prior to their enrollment in the study. For children (age < 18 years), consent was obtained from the parent prior to data collection.

## 5. Conclusions

In conclusion, there are concerns regarding the appreciable dispensing of antimicrobials among drug sale outlets in Pakistan, including for patients with COVID-19, alongside the appreciable dispensing of ‘Watch’ versus ‘Access’ antibiotics. In addition, the majority of the drug sale outlets in Punjab were working without the presence of a community pharmacist or pharmacy technicians where antibiotics were being dispensed without a prescription. This needs to be urgently addressed going forward. This includes greater education of community pharmacists, pharmacy technicians and the public.

ASPs can successfully be implemented in these drug sale outlets along with improved education among the dispensers and patients to reduce inappropriate requests and dispensing of antibiotics without a prescription, and hence reduce AMR. Subsequently, regular monitoring of the interventions among sales outlets is necessary to continue to reduce inappropriate dispensing. This is important to meet key goals of the current NAP on AMR. 

## Figures and Tables

**Figure 1 antibiotics-12-01018-f001:**
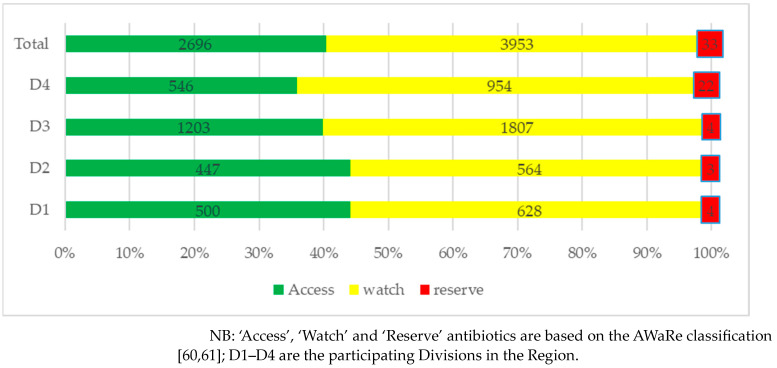
Dispensing practices of antibiotics according to AWaRe classification distributed by Division.

**Figure 2 antibiotics-12-01018-f002:**
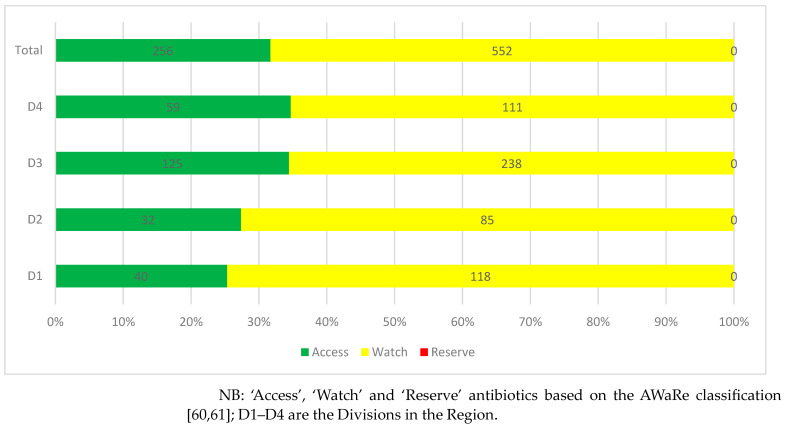
Dispensing practices of antibiotics among patients with COVID-19 distributed by Division.

**Table 1 antibiotics-12-01018-t001:** Information about the drug sale outlets and encounters.

Division	D1		D2		D3		D4		Total (%)
Drug Outlets	Pharmacy	Medical Store	Pharmacy	Medical Store	Pharmacy	Medical Store	Pharmacy	Medical Store	
Numbers of drug sale outlets per division	9	13	8	10	12	16	10	14	92
**Drug outlet type**									
Chain	4	0	5	2	5	4	4	2	26 (28.3)
Independent	5	13	3	8	5	12	3	12	66 (71.7)
**Location of drug sale outlets**									
Urban	7	6	7	5	10	6	8	7	56 (60.9)
Rural	2	7	1	5	2	10	2	7	36 (39.1)
**Presence of qualified person**									
Yes	3	3	5	3	5	3	5	3	30 (32.6)
No	6	10	3	7	7	13	5	11	62 (67.4)
No. of encounters	1322	812	1405	736	2673	1436	1792	916	11,092
**Gender**									
Male	872	455	752	245	1671	652	967	507	6121 (55.2)
Female	450	357	653	491	1002	784	825	409	4971 (44.8)
**Age (years)**									
<15	324	136	371	152	581	354	418	171	2507 (22.6)
15–30	112	115	161	73	365	206	247	137	1416 (12.8)
31–45	267	110	246	118	472	275	308	224	2020 (18.2)
46–60	362	234	340	236	893	414	512	287	3278 (29.5)
>60	257	217	287	157	362	187	307	97	1871 (16.9)

NB: D1–D4 refer to the Divisions in the Region.

**Table 2 antibiotics-12-01018-t002:** Detail of dispensed antimicrobials.

Regions	D1	D2	D3	D4	Total (%)
Total encounters	2134	2141	4109	2708	11,092
Encounters supplied with an antimicrobial	1252	1446	2937	1810	7445 (67.1)
**Type of antimicrobial dispensed**					
Antibiotics	1132	1014	3014	1522	6682 (74.3)
Antivirals	43	26	235	252	556 (6.2)
Antifungals	87	112	442	273	914 (10.2)
Anthelmintics	126	171	311	106	714 (7.9)
Antiprotozoals	12	31	54	34	131 (1.4)
Total	1400	1354	4056	2187	8997
**Encounters suppled with an antimicrobial by route of administration**					
Oral	640	720	2168	854	4382 (48.7)
IV	703	613	1562	1172	4050 (45.1)
Topical	57	21	326	161	565 (6.2)
**Antimicrobials supplied on prescription**					
Yes	940	795	2847	1423	6005 (66.8)
No	460	559	1209	764	2992 (33.2)

NB: D1–D4 refer to the Divisions in the Region.

**Table 3 antibiotics-12-01018-t003:** Indications for dispensed antibiotics.

Indications	D1	D2	D3	D4	Total (%)
Respiratory tract infection	382	316	1072	523	2293 (34.3)
Gastrointestinal infection	133	177	526	284	1120 (16.8)
Skin and soft-tissue infection	162	153	367	178	860 (12.9)
Pre- or post-operative prophylaxis	171	132	381	158	842 (12.6)
COVID-19	158	117	363	170	808 (12.0)
Urinary tract infection	76	68	133	136	413 (6.2)
Eye/ear infection	36	24	118	41	219 (3.3)
Others	14	27	54	32	127 (1.9)

NB: D1–D4 refer to the participating Divisions in the Region.

**Table 4 antibiotics-12-01018-t004:** Top ten most frequently dispensed antibiotics from drug sale outlets.

Name of the Antibiotic and Classification	D1	D2	D3	D4	Total (%)
Ceftriaxone—‘W’	226	148	621	234	1229 (18.4)
Amoxicillin—‘A’	213	208	426	186	1033 (15.4)
Azithromycin—‘A’	156	157	474	211	998 (14.9)
Metronidazole—‘A’	122	96	305	126	649 (9.7)
Ciprofloxacin—‘A’	111	131	265	123	630 (9.4)
Co-amoxiclav—‘A’	86	66	183	84	419 (6.3)
Cefixime—‘W’	58	46	137	68	309 (4.6)
Cefoperazone + sulbactam—‘W’	43	28	126	48	245 (3.7)
Doxycycline—‘A’	36	23	86	36	181 (2.7)
Co-trimoxazole—A’	18	13	98	27	156 (2.3)

NB: ‘A’ and ‘W’—AWaRe classification [60,61]. D1–D4 refer to participating Divisions in the Region.

**Table 5 antibiotics-12-01018-t005:** Dispensed antibiotics for COVID-19.

Name of Agent	D1	D2	D3	D4	Total (%)
Ceftriaxone—‘W’	34	27	114	52	227 (28.0)
Azithromycin—‘W’	53	32	78	39	202 (25.0)
Amoxicillin—‘A’	19	16	103	43	181 (22.4)
Ciprofloxacin—‘W’	18	15	35	20	88 (10.9)
Co-amoxiclav—‘A’	21	16	22	16	75 (9.2)

NB: ‘A’ and ‘W’—AWaRe classification [60,61]. D1–D4 refer tothe participating Divisions in the Region.

## Data Availability

Further additional data are available from the corresponding authors on reasonable request.

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
