# Peer review of "Antimicrobial Dispensing Practices during COVID-19 and the Implications for Pakistan"

_antibiotics, 2023, doi:10.3390/antibiotics12061018_

Round 1

Reviewer 1 Report

Dear authors, 

I reviewed this paper and have the following comments:

MAJOR COMMENTS

-line 28-29: it is misuse and abuse of antibiotics that increase AMR. Please amend.

-the whole introduction needs to be rewritten in my view. It is not clear why it is possible to buy antibiotics without prescription in a country with high prevalence of MDR/XDR pathogens that seems to have a NAP addressing AMR. In addition the study is only briefly described in the abstract. 

-line 49: COVID-19 is a pandemic, not an outbreak

-line 51: SARSCoV2 is the virus, COVID-19 is the disease. I assume what is reported is the first case with a SARSCoV2 infection. If so please amend

-line 60: please replace currently with the time period when this is reported

-line 68: what was the prevalence of coinfections and secondary bacterial infections in Pakistan? Please spell out.

-lines 107-108: what is the difference between a medical store and a pharmacy? Not all readers are familiar with how antibiotics can be dispensed/purchased in Pakistan. This should be explained. Are all antibiotics prescription-only medicines? Are any of them OTC and which? How many of the pharmacies were hospital pharmacies?

-line 110: please replace customers with patients

-what regions are D1 to D4 and why are they not spelled out?

-the discussion states that many patients received more than 1 antibiotic. What were the combinations prescribed and for which indication?

-the discussion repeats some of the results without attempting to discuss/explain them. This needs to be reconsidered.

MINOR COMMENTS

-line 30-31: the sentence starting with "consequently" does not contain a verb; please have a look.

-line 76: replace dipropionate with disproportionate

There are some typos and unfinished phrases in the text and the style needs to be improved.

Author Response

Comments and Suggestions for Authors

Dear authors, 

I reviewed this paper and have the following comments

Author comments: Thank you for this – we hope we have addressed your issues and concerns where we can.

  1. A) MAJOR COMMENTS

1) -line 28-29: it is misuse and abuse of antibiotics that increase AMR. Please amend.

Author comments: Thanks you – now amended

2) the whole introduction needs to be rewritten in my view. It is not clear why it is possible to buy antibiotics without prescription in a country with high prevalence of MDR/XDR pathogens that seems to have a NAP addressing AMR. In addition the study is only briefly described in the abstract. 

Author comments: Thank you for this comment. We have now updated this section. As you can see, unfortunately self-purchasing is prolific across a number of LMICs including Asia and Africa despite very high rates of AMR (as seen in e.g. Africa). This is why in the Discussion and Supplementary Material we have documented activities that have reduced inappropriate self-purchasing – as well as where this has not worked well and why - to give future guidance to all key stakeholder groups in Pakistan as the country moves forward with its NAP. We hope this is now OK.

3) line 49: COVID-19 is a pandemic, not an outbreak

Author comments: Thank you – addressed.

4) line 51: SARSCoV2 is the virus, COVID-19 is the disease. I assume what is reported is the first case with a SARSCoV2 infection. If so please amend

Author comments: Thank you – now amended.

5) line 60: please replace currently with the time period when this is reported

Author comments: Thank you. We have updated it.

6) line 68: what was the prevalence of coinfections and secondary bacterial infections in Pakistan? Please spell out.

Author comments: Thank you - now updated.

7) lines 107-108: what is the difference between a medical store and a pharmacy? Not all readers are familiar with how antibiotics can be dispensed/purchased in Pakistan. This should be explained. Are all antibiotics prescription-only medicines? Are any of them OTC and which? How many of the pharmacies were hospital pharmacies?

Author comments: Thank you – now updated. The Drug Laws in Pakistan state that antibiotics should only be dispensed and sold with a prescription (now made clearer). However, lax enforcement has resulted in high rates of self-purchasing as seen. There were no hospital pharmacies in this study – with patients often requested to purchase antibiotics from community pharmacies if stocks are low in hospital.

8) line 110: please replace customers with patients

Author comments: Thank you – now addressed.

9) what regions are D1 to D4 and why are they not spelled out?

Author comments: Thank you – this is short hand for the 4 Districts conveniently selected for this study rather than actually spelling them out – we hope this is OK.

10) the discussion states that many patients received more than 1 antibiotic. What were the combinations prescribed and for which indication?

Author comments: Thank you. Many patients received more than one antibiotic. However, we didn’t collect the information separately for specific antibiotics against a particular indication if more than one. We just collected details of all antibiotics dispensed for a particular indication whether it was one or two antibiotics for that indication in a given patients in line with our objectives for the study. We have mentioned this as a limitation and hope this is acceptable.

11) the discussion repeats some of the results without attempting to discuss/explain them. This needs to be reconsidered.

Author comments: Thank you for this. As seen, we have updated the Discussion. We have now also added in (and expended on this) regarding the implications for key stakeholders in Pakistan going forward including the purchasing of antibiotics without a prescription – especially where initiatives have not been successful and why (The Supplementary Tables have now been updated to include this new information – and we are aware these Tables were missed out from the initial paper sent to you). We hope this is now OK.

  1. B) MINOR COMMENTS

1) line 30-31: the sentence starting with "consequently" does not contain a verb; please have a look.

Author comments: Thank you – now addressed.

2) line 76: replace dipropionate with disproportionate

Author comments: Thank you – now addressed.

  1. C) Comments on the Quality of English Language

There are some typos and unfinished phrases in the text and the style needs to be improved.

Author comments: Thank you – now addressed. As previously mentioned, one of the co-authors has over 500 publications in peer-reviewed Journals - and we trust this is now acceptable.

Reviewer 2 Report

In this manuscript the authors have studied the antimicrobial dispensing practices during COVID-19 in Pakistan. They undertook this study as understanding what antibiotics are prescribed for what conditions is crucial in reducing AMR.

Line 43: "reduce" AMR

Line 52: Please confirm the date of first positive case of COVID-19. Feb, 2022 appears wrong.

Line 53-57: Irrelevant to the topic

Line 76: Disproportionate

Line 107: Reformat 92.

Table 2 needs reformatting. Because the table is divided in two pages, the missing headings on Page 4 makes it hard to comprehend.

Table 2 : Please explain " Encounters suppled"

Line 121: Needs reformatting

English language is fine, other than some minor reformatting of the sentences as mentioned in the comments.

Author Response

Comments and Suggestions for Authors

In this manuscript the authors have studied the antimicrobial dispensing practices during COVID-19 in Pakistan. They undertook this study as understanding what antibiotics are prescribed for what conditions is crucial in reducing AMR.

Author comments: Thank you for this summary – appreciated!

1) Line 43: "reduce" AMR

Author comments: Thank you – now addressed.

2) Line 52: Please confirm the date of first positive case of COVID-19. Feb, 2022 appears wrong.

Author comments: Thank you – now addressed.

3) Line 53-57: Irrelevant to the topic

Author comments: Thank you – we have cut down on some of this. However, we have retained some elements as we believe this helps set the scene for the paper. We hope this is acceptable.

4) Line 76: Disproportionate

Author comments: Thank you – now addressed.

5) Line 107: Reformat 92.

Author comments: Thank you – now addressed.

6) Table 2 needs reformatting. Because the table is divided in two pages, the missing headings on Page 4 makes it hard to comprehend.

Author comments: Thank you – now addressed.

7) Table 2 : Please explain " Encounters suppled".

Author comments: Encounter means the appearance of a patient/customer obtaining a medicine form a drug sale point. Encounters that were dispensed antibiotics were subsequently named as encounters supplied with antibiotics. We have now added in more details to the paper and trust this is now OK.

8) Line 121: Needs reformatting

Author comments: Thank you – now addressed.

9) Comments on the Quality of English Language

English language is fine, other than some minor reformatting of the sentences as mentioned in the comments.

Author comments: Thank you. As mentioned, we have now been through the manuscript with the help of one of the co-authors with over 500 publications in peer-reviewed Journals and we trust this is now acceptable.

Round 2

Reviewer 1 Report

Thanks for the provided response. My comments were addressed. One additional point: please change the title of figure 2. Antibiotics are not prescribed against COVID-19.

Some examples where English could be improved.

Lines 51-52: since the emergence of the COVID-19 outbreak caused by SARS-COV2 outbreaking December 2019. .

Lines 89-90: globally in 2019 [...], with this figure raising. Compared to what?

Lines 224-225: [...]antibiotics dispensed in our study[...]. How about "antibiotics dispensed by the drug outlets included in our study"?

Lines 254-255:  We have seen trained pharmacists, coupled with 254 guidelines suggesting alternatives to antibiotics, being used to good effect in LMICs

Author Response

Comments and Suggestions for Authors

1) Thanks for the provided response. My comments were addressed. One additional point: please change the title of figure 2. Antibiotics are not prescribed against COVID-19.

Author comments: Thank you for this – now addressed.

2) Comments on the Quality of English Language: Some examples where English could be improved.

Author comments: Thank you – we have now been through the paper and further updated it. We trust this is now acceptable.

3) Lines 51-52: since the emergence of the COVID-19 outbreak caused by SARS-COV2 outbreaking December 2019.

Author comments: Thank you – now amended

3) Lines 89-90: globally in 2019 [...], with this figure raising. Compared to what?

Author comments: Thank you – now amended

4) Lines 224-225: [...]antibiotics dispensed in our study[...]. How about "antibiotics dispensed by the drug outlets included in our study"?

Author comments: Thank you – now amended

5) Lines 254-255:  We have seen trained pharmacists, coupled with 254 guidelines suggesting alternatives to antibiotics, being used to good effect in LMICs

Author comments: Thank you – now amended

Reviewer 2 Report

Thanks for addressing the concerns.

Author Response

Comments and Suggestions for Authors

Thanks for addressing the concerns.

Author comments: Thank you also for your help -appreciated!
